# Serine Hydrolases in Lipid Homeostasis of the Placenta-Targets for Placental Function?

**DOI:** 10.3390/ijms23126851

**Published:** 2022-06-20

**Authors:** Natascha Berger, Hanna Allerkamp, Christian Wadsack

**Affiliations:** 1Department of Obstetrics and Gynecology, Medical University of Graz, 8036 Graz, Austria; natascha.berger@medunigraz.at (N.B.); hanna.allerkamp@medunigraz.at (H.A.); 2BioTechMed-Graz, 8036 Graz, Austria

**Keywords:** serine hydrolases, lipases, placenta, placental metabolism, pregnancy disorders

## Abstract

The metabolic state of pregnant women and their unborn children changes throughout pregnancy and adapts to the specific needs of each gestational week. These adaptions are accomplished by the actions of enzymes, which regulate the occurrence of their endogenous substrates and products in all three compartments: mother, placenta and the unborn. These enzymes determine bioactive lipid signaling, supply, and storage through the generation or degradation of lipids and fatty acids, respectively. This review focuses on the role of lipid-metabolizing serine hydrolases during normal pregnancy and in pregnancy-associated pathologies, such as preeclampsia, gestational diabetes mellitus, or preterm birth. The biochemical properties of each class of lipid hydrolases are presented, with special emphasis on their role in placental function or dysfunction. While, during a normal pregnancy, an appropriate tonus of bioactive lipids prevails, dysregulation and aberrant signaling occur in diseased states. A better understanding of the dynamics of serine hydrolases across gestation and their involvement in placental lipid homeostasis under physiological and pathophysiological conditions will help to identify new targets for placental function in the future.

## 1. Introduction

The group of serine hydrolases represents one of the largest and most diverse classes of hydrolytic enzymes, with a broad range of substrates. Ubiquitously expressed, these proteins possess the ability to catalyze the hydrolysis, as well as the synthesis, of esters. The common mechanism for this water-induced reaction is the reversibly cleavage of covalent bonds, thereby dividing large molecules into active, smaller breakdown products, making them available as substrates for various downstream processes. In general, hydrolases function as important contributors to virtually all physiological and pathophysiological events in mammals, including neuronal processes, metabolism, reproduction, inflammation, and cancer [1,2,3,4,5]. Examples of common hydrolases include esterases, proteases, glycosidases, nucleosidases, and lipases. Lipid-specific hydrolases catalyze the hydrolysis of more complex lipids into simpler lipids. For example, lipases break ester bonds of water-insoluble long-chain fatty acids (C > 12), and carboxylesterases hydrolyze esters of short to medium-chain carboxylic acids (C ≤ 12) [6]. Fatty acids and lipids play important roles throughout pregnancy, representing key elements for metabolic processes and energy production to support the development and growth of the fetus and the placenta. Both are provided and metabolized through the action of lipid hydrolases, which, e.g., determine the release of fatty acids from lipid droplets and membrane or lipoprotein-associated lipids. The expression and activity of placental lipid hydrolases are influenced by various factors, such as the gestational age, diet, and state of health of the mother [7,8,9]. This review focuses on lipid-metabolizing serine hydrolases, with emphasis on the enzymes in female reproductive tissues and their impact on placental function, which thereby contribute to fetal development in healthy and compromised pregnancies (Table 1).

### 1.1. The Metabolically Active Human Placenta

The human placenta (Figure 1) serves as a dynamic contact zone between the mother and the fetus. It plays a central metabolic role in pregnancy by fulfilling the biosynthetic requirements that support its own quick growth and that of the fetus. The placental multicellular barrier can be morphologically segmented into the fetal-facing chorionic plate, from which the umbilical cord originates; the maternal-facing basal plate, composed of the innermost lining of the uterus (decidua); and invading placental cells. Out of the chorion, placental villi arise, which bathe in the maternal blood filling the intervillous space. The tree-like villous structure constitutes the unique area for maternal–fetal exchange and its outermost layer is composed of the epithelial-like trophoblast cells, which constitute the first barrier facing the maternal blood. The multinucleated syncytiotrophoblast (STs) originates from the underlying proliferating and differentiating cytotrophoblasts (CTs). The villous stroma is composed of connective tissue occupied by placental macrophages, the so-called Hofbauer cells (HBCs), fibroblasts, and feto-placental blood vessels fitted in between [10,11].

### 1.2. Characterization of Lipid Hydrolases

Lipid hydrolases are classified according to their structure, function, and substrate specificity. The Lipase Engineering Database (https://led.biocatnet.de/, accessed on 9 May 2022) so far offers approximately 280,000 protein sequences and 1600 protein structures in total, highlighting the importance of integrating the sequence, structure, and biological function of lipases. Structural analysis has revealed that the majority of these enzymes are based on an α/β hydrolase fold, which generally consists of eight mostly parallel β-sheets surrounded by six α-helices, with some variations among several lipases [12,13]. Phospholipase A2 and the amidase enzymes encompass structurally distinct members of the α/β hydrolase superfamily [14,15]. Other common structural features for most lipid hydrolases are the highly conserved catalytic triad embedded within the consensus pentapeptide sequence GXSXG [16]. The catalytic triad harbors a serine as the nucleophile, a basic histidine, and an acidic aspartate or glutamate. Additional modules of α/β hydrolases are a lid, a cap, and an N-terminal or C-terminal domain [17,18]. Despite the conserved catalytically active sites in one α/β hydrolase family, those enzymes display reactions on a wide variety of substrates and are reported to exert catalysis-independent functions, such as protein–protein interactions [19]. The catalytic mechanism of hydrolases is initiated by the formation of an acyl-enzyme intermediate, which activates the hydroxyl group of the catalytic serine, leading to increased nucleophilicity. The active-site amino acids form the so-called oxyanion hole that stabilizes the reaction intermediate. Consequently, the carbonyl group of the substrate is attacked, and the product is released through a water-induced deacylation, thereby regenerating the enzyme for subsequent reactions [16].

## 2. Hydrolase Types and Implications in Placental Biology

The human serine hydrolase family encompasses more than 200 different enzymes, which have been classified according to the topological site of enzymatic action and/or their substrate specificity (Figure 2).

### 2.1. Intracellular Neutral Lipases

The catabolism of neutral lipids, such as triglycerides (TGs) and cholesteryl esters (CEs), in adipose and non-adipose tissues is a complex and hormonally regulated process. If energy is required, TGs stored in lipid droplets within cells are hydrolyzed, and the generated fatty acids are either oxidized in the mitochondria, producing energy, or act as bioactive signaling molecules. The regulation of lipolysis is accomplished by several serine hydrolases. In this section, enzymes that are reported to be expressed in human placental tissue, including adipose triglyceride lipase (ATGL), lysosomal acid lipase (LAL), hormone-sensitive lipase (HSL), diacylglycerol lipase α and β (DAGLα/β), monoacylglycerol lipase (MGL), and the α/β-hydrolase domain (ABHD)-containing proteins ABHD6 and ABHD12, are described (Figure 2).

#### 2.1.1. Adipose Triglyceride Lipase (ATGL)

Human ATGL, also known as patatin-like phospholipase domain containing 2 (PNPLA2), is a ~55 kDa protein that catalyzes the first step in TG hydrolysis at the surface of cytosolic lipid droplets. Importantly, CGI-58, also called ABHD5, is required as co-activator for hydrolytic catalysis [20,21,22]. ATGL is characterized by a catalytical dyad bearing the active-site serine and an aspartate, located in the patatin domain [23,24]. Various studies looking at pharmacological inhibition of ATGL or total knockout of the gene in mice showed an increase in the adipose tissue mass and size of lipid droplets, as well as alterations in energy metabolism [21,23,25]. Mutations in the gene coding for ATGL lead to neutral lipid storage disease with or without myopathy [23]. ATGL mRNA and protein have been detected in placental tissue and predominantly localized to the ST, with less staining of the endothelium, stroma (including HBCs), and decidua, as shown by immunohistochemistry [26,27]. ATGL mRNA levels correlate with maternal pre-pregnancy BMI and transcription of ATGL was elevated in placentae of women who suffered from gestational diabetes mellitus (GDM) [26,28]. From these descriptive studies, one may speculate that maternally derived metabolic derangements alter ATGL function in the placenta, thereby contributing to the well-described dyslipidemia in placental tissue.

#### 2.1.2. Hormone-Sensitive Lipase (HSL)

HSL or LIPE (84 kDa) is a cytosolic and ubiquitously expressed enzyme with the highest expression in white and brown adipose tissue [29]. HSL exerts its hydrolase activity with a broad range of substrates, including TGs, diglycerides (DGs), monoglycerides (MGs), CEs, and retinol ester [30,31]. Importantly, genetic deletion of HSL in mice led to a drastic DG accumulation in various tissues [29]. Furthermore, in vitro studies demonstrated that HSL predominantly hydrolyzes DGs over TGs and MGs and shows sn-3 stereo selectivity [32]. Anti-/lipolytic hormones, depending on the nutritional state, regulate the activity of HSL. Studies in rodents revealed that HSL is phosphorylated by protein kinase A in response to β-adrenergic stimulation during periods of fasting, whereas insulin inhibits HSL activity during feeding. The moderate induction of HSL is substantially increased by translocation of the enzyme from the cytosol to the surface of lipid droplets. Subsequently, hydrolysis is initiated by binding of HSL to the lipid droplet-associated protein perilipin, where it exerts its full activity [30,33,34]. Barrett et al. investigated mRNA and protein expression of HSL in normal term placentae and compared the expression of HSL to GDM and preeclamptic placental tissues. Immunohistochemical analysis localized HSL to the ST, endothelium, HBCs, and decidual cells. The same study found reduced HSL expression in placentae of preeclamptic women [26,27]. Another study showed an association of elevated placental HSL expression with maternal type 1 diabetes [35]. Although, again, only descriptive studies on human tissue are available, the current knowledge indicates that aberrant placental HSL expression is associated with metabolic diseases in the mother.

#### 2.1.3. Diacylglycerol Lipase α and β (DAGLα/β)

The products of the two closely related genes DAGLα and DAGLβ share about 30% homology and are composed of four transmembrane domains at the N-terminus followed by the catalytic domain, which contains the nucleophilic serine residue. Both enzymes, DAGLα (115 kDa) and DAGLβ (74 kDa), are found in various species and show a strong homology between human and mouse [36]. It has been shown that glutathione and Ca2+ stimulate the activity of both enzymes [36], whereas calcium/calmodulin-dependent protein kinase II inhibits mouse DAGLα in the brain by phosphorylation of two serine residues at the C-terminal domain [37]. DAGLα/β possess sn-1-specific hydrolytic activity for arachidonic acid (AA)-esterified DGs, thereby generating 2-arachidonoylglycerol (2-AG), which is one of the main endocannabinoids and the most abundant ligand for cannabinoid receptors type 1 and 2 [38]. 2-AG, as the precursor of AA, is essentially involved in fetal neurodevelopment and axonal guidance [39]. Recently, Shin et al. identified TG lipase activity on DAGLβ. Notably, DAGLβ showed robust hydrolysis activity for triarachidonin (C20:4 FA) and tridocosahexaenoin (C22:6 FA) in vitro [40]. Interestingly, both DAGL isoforms exhibit diverse cell-type and tissue-specific abundance. The DAGLα isoform is expressed in tissues of the central nervous system and is particularly enriched in neurons. In line with this, lipid analysis of DAGLα knockout mice demonstrated at least 80% reduction of 2-AG and AA levels in the brain [41]. Apart from its pivotal role in 2-AG metabolism in the brain, DAGLα immunostaining in pregnant murine uteri showed a spatiotemporal differential expression of this enzyme in early pregnancy. On day four prior to implantation, immunostaining for DAGLα revealed high expression at the apical luminal epithelium and myometrium. Interestingly, on days five to seven of implantation, DAGLα expression was increased at the inter-implantation site [42]. These results indicate tight regulation of 2-AG synthesis at sites of implantation. In human reproductive tissues, immunohistochemical analysis of term placentae localized DAGLα to the CTs and STs [43,44,45]. Recently, the impact of tetrahydrocannabinol (THC) on DAGLα mRNA and protein expression was investigated using placental explants. After 24 h of treatment with the highest concentration of THC (40 µM), significantly elevated DAGLα levels were detected, suggesting that THC induces key enzymes of the endocannabinoid system in vitro [45]. In contrast, DAGLβ is mainly expressed in peripheral tissues such as in the liver, where DAGLβ knockout mice showed 90% reductions of 2-AG levels [38]. Compared to DAGLα, DAGLβ activity is enriched in immune cells including microglia [41], macrophages [46] and dendritic cells [47]. It was reported that microglia of DAGLβ depleted mice showed a 50% reduction in 2-AG, AA and prostaglandin levels compared to wild type mice [41]. However, data on the mRNA expression of placental DAGLβ was only shown in baboon and rat placentae in the context of maternal obesity and high omega-6 linoleic acid dietary intake, respectively [48,49]. Our preliminary data revealed that the number of DAGLβ transcripts was significantly elevated, compared to DAGLα, in human term placental tissues. Furthermore, DAGLβ transcripts were clearly located to the CK7 positive ST, but could hardly be assigned to the endothelium or HBCs. In contrast, DAGLα transcripts were detected in all three cell types although in lower abundancies. To our knowledge, we detected DAGLβ activity in human placentae for the first time and could show that pharmacological inhibition of the enzyme led to a significant decrease in tissue 2-AG levels. Our first results suggest that DAGLβ regulates 2-AG supply in the placenta, which opens new considerations on the importance of the enzyme in AA supply and moreover related mechanisms such as prostanoid synthesis.

#### 2.1.4. Monoacylglycerol Lipase (MGL)

Human MGL (33.4 kDa) shows a sequence homology of 83.8% to mouse MGL. In mice, this enzyme is ubiquitously expressed but the highest expression is reported in the brain, white adipose tissue, and liver [50,51]. The protein’s crystal structure shows that the soluble enzyme has an amphipathic character thanks to its hydrophobic lid domain, which allows the association of MGL with cellular membranes, harboring its lipophilic substrates [52]. MGL preferentially hydrolyzes MGs compared to DGs and TGs and is specifically involved in the degradation of the most abundant endocannabinoid 2-AG [51]. Selective inhibition and genetic disruption of MGL led tosignificantly reduced AA and downstream AA-derived eicosanoid levels in central and peripheral tissues of mice and demonstrated that MGL determines 2-AG signaling [53]. Moreover, it has been shown that inhibition of MGL has neuroprotective effects in animal models of Parkinson disease, acute brain injury, and multiple sclerosis and, currently, a selective drug candidate is entering clinical phase 2 studies for neurological disorders [54,55]. Protein expression of MGL in isolated CTs, BeWo cells [43], and placental tissue sections [44] was described. Costa et al. investigated the effect of 2-AG on cell proliferation/viability and morphology and examined the underlying molecular pathways. They showed that 2-AG induced, time- and dose-dependently (10–25 µM), a decrease in cell viability, reduced DNA synthesis, and BeWo cells displayed morphological patterns of apoptosis after treatment. Conversely, treatment with 2-AG in CT had no effect on cell viability or cytotoxicity. It was further shown in CT that high concentrations of 2-AG (10, 20 µM) decreased placental alkaline phosphatase activity and human chorionic gonadotropin secretion significantly [43,44]. Moreover, the same group detected MGL expression in the ST layer of placental tissue sections and showed that THC impairs mRNA and protein expression of MGL in chorionic villi explants in a time-dependent manner [45]. Furthermore, a study in rat placentae showed that MGL gene expression correlates with gestational age from mid-late gestation to onset of labor, indicating an increase in 2-AG hydrolysis and production of AA for downstream prostaglandin synthesis [56]. Interestingly, MGL is differently expressed in murine uteri during the process of implantation. These data suggest a critical role for MGL in regulating 2-AG levels at the implantation site to protect the embryo in early stages of development [42]. Moreover, Guida et al. investigated aberrant expression of the endocannabinoid system in endometrial biopsies of patients suffering from endometrial cancer and revealed that MGL protein expression was significantly decreased compared with healthy patients [57]. This finding was in line with elevated 2-AG and cannabinoid receptor 2 levels in endometrial carcinoma tissues [57]. The presence and dynamics of the two key enzymes, DAGLβ and MGL, involved in 2-AG metabolism in human reproductive tissues provide support for the importance of endocannabinoid signaling during pregnancy and for the hypothesis that its dysregulation may be at least partly responsible for altered placental development and poor pregnancy outcomes.

#### 2.1.5. α/β-Hydrolase Domain Containing Proteins ABHD6 and ABHD12

Human ABHD6 (38 kDa) and ABHD12 (45 kDa) are hydrolases with a single transmembrane domain that are located in the cytosol or, in the case of ABHD12, in the lumen of the endoplasmic reticulum or extracellular space [58,59]. Although MGL predominantly regulates 2-AG abundance in the central nervous system, both ABHD enzymes also show 2-AG hydrolase activity in vitro, differing from each other in their substrate isomer specificity [60,61]. ABHD6 is ubiquitously expressed in multiple human tissues, including the brain, lungs, and liver [59,62]. ABHD12 expression is mainly characterized by the neurodegenerative disease called polyneuropathy, hearing loss, ataxia, retinitis pigmentosa, and cataract (PHARC) syndrome, which is associated with mutations and deletions in the Abhd12 gene in humans and might be a consequence of a defective endocannabinoid system [63]. Recently, ABHD6 and ABHD12 mRNA and protein were reported in term placental explants. The treatment of explants with THC induced an elevation of ABHD6/12 mRNA and protein levels in a time- and concentration-dependent manner. ABHD6/12 expression increased after a continuous THC stimulus for 72 h, while MGL expression decreased after an initial peak at 24 h [45]. These data are in concordance with previous findings demonstrating that ABHD6 activity fine-tunes 2-AG signaling, suggesting specialized roles for these enzymes in the placental endocannabinoid system [60].

#### 2.1.6. Lysosomal Acid Lipase (LAL)

Lysosomal acid lipase (LAL), encoded by the lipase A gene (LIPA), is an intracellular active glycoprotein in humans with a molecular weight of about 40 kDa. LAL mRNA is highly expressed in the brain, lungs, kidneys, and, to a smaller extent, in the placenta, liver, and heart [64,65]. This enzyme shows maximal activity at an acidic pH (pH ~ 4.160) against CE, TGs, DGs, and MGs in vitro and is localized in early and late endosomal membranes. LAL hydrolyzes CEs from internalized LDL particles, thereby supplying the cell with cholesterol [66,67]. It has been reported that adult LAL-depleted mice show massive accumulations of TGs and CEs in the liver, adrenal glands, and small intestines [68]. A complete loss of LAL activity in humans is manifested as Wolman’s disease. This disease, which in the worst case can lead to the patient’s death within the first year of life, is characterized by intra-lysosomal TG and CE accumulation in many tissues and is expressed by vomiting, diarrhea, and hepatosplenomegaly. Patients maintaining residual LAL activity are affected by cholesteryl ester storage disease, which is characterized by hypercholesterolemia [69,70]. The mRNA expression of LAL was examined in tissue biopsies of human placentae and showed the highest copy number compared to other TG-hydrolyzing enzymes [35]. Recently, a study investigated the effect of genetic disruption of murine LAL and demonstrated CE accumulation in placentae and fetuses, suggesting that lysosomal rather than neutral lipolysis already alters placental and fetal cholesterol homeostasis *in utero*. Furthermore, LAL deficiency led to massive hepatic lysosomal lipid accumulation after birth, with a severe progression in young adulthood [71]. The insight that aberrant LAL expression leads to a severe metabolic disorder not only during pregnancy but also in the first weeks of life in mice underlines the crucial role of this hydrolase in lysosomal CE metabolism. Consequently, more studies in human reproductive tissues examining the function of LAL would be of great value.

### 2.2. Extracellular Lipases

Members of the extracellular TG lipase gene family, including hepatic lipase, endothelial lipase (EL), lipoprotein lipase (LPL), and pancreatic lipase, are capable of releasing fatty acids originating from either TGs or phospholipids (PLs) containing lipoproteins. There are mainly two lipases described in the human placenta, acting at the maternal–fetal interface by providing fatty acids for placental uptake (Figure 2) [9].

#### 2.2.1. Lipoprotein Lipase (LPL)

LPL possesses a variety of substrate activities; it hydrolyzes fatty acids from TG-enriched very-low-density lipoproteins (VLDLs) but also from PL-enriched high-density lipoproteins (HDL) [9]. LPL (~55 kDa) is a glycoprotein that is primarily expressed by STs, stroma cells, and HBCs [27,72]. The physiologically active form of LPL exists as a homodimer and requires apolipoprotein C-II as co-activator, which is a component of HDLs and VLDLs [2,73]. In adipose tissue, LPL is secreted by parenchymal cells and transported to the luminal surface of the vascular endothelium, where it binds to heparin sulfate proteoglycans of capillary endothelial cells [73]. Lindegaard and co-workers detected LPL mRNA predominantly in the syncytium of placental tissues, whereas immunohistochemical staining revealed that the protein was located in trophoblast and endothelial cells. They suggest that LPL may be translocated from trophoblast cells to the vascular lumen of the placenta in analogy to LPL in adipose tissue or that it derives from fetal blood [72]. LPL activity has been assessed in placental microvillus membranes [74], isolated trophoblast cells, and HBCs [75], and it is thought to be tightly regulated, since it is involved in the initial step of transplacental fatty acid transport. Regulation of enzyme activity can occur at multiple levels, including gene expression, intracellular transport and secretion, glycosylation, dimerization and degradation, that concern varying physiological states linked to specific metabolic demands [2,73]. In this context, it has been shown that LPL activity increases across gestation, with a threefold higher activity at term compared to first trimester villous tissue. Furthermore, LPL activity was stimulated by insulin and glucose and can additionally be regulated by hormones such as cortisol, IGF−1, and estradiol in term placental villous tissue [76]. In contrast, a downregulation of LPL in GDM- and type 1 diabetic-complicated pregnancies was detected by transcriptome profiling [77], underlining the discrepancy between in vitro and in situ studies. Furthermore, LPL activity in isolated trophoblast cells was reduced by high levels of maternal TG and/or FFA, which might counteract excessive uptake and delivery of FFA to the fetus [78]. Moreover, in intrauterine growth-restriction (IUGR), an increase of placental LPL mRNA expression was observed, indicating altered placental fatty acid homeostasis [79,80,81]. Conversely, it was reported that LPL contributes to fetal fat accretion through increased LPL activity that was detected in placental villous tissue, linked to newborn adiposity [82]. The ambiguity of the current literature emphasizes the complex regulation of this enzyme, which presumably depends on the metabolic state and demand of both mother and fetus. More detailed studies on the contributions of LPL to the specific mechanisms involved in the pathogeneses of different pregnancy complications are needed.

#### 2.2.2. Endothelial Lipase (EL)

Human EL is encoded by the LIPG gene and shares about ~30–40% homology with LPL and hepatic and pancreatic lipase, all members of the TG lipase gene family [83]. EL is primarily synthesized and secreted by vascular endothelial cells and subsequently modified by glycosylation, generating a 68 kDa protein. Like LPL, EL binds to proteoglycans after secretion on the luminal surface of the vascular endothelium [83,84]. EL is mainly expressed in tissues such as the liver, lung, kidney, thyroid, ovary, testis, and placenta [83]. Analysis of human placental tissue and isolated primary cells revealed that EL mRNA and concomitant protein expression is localized to both ST and endothelial cells [72,81]. It was demonstrated that EL acts predominantly as a sn−1-specific phospholipase but is also capable of hydrolyzing short- and long-chain fatty acyl groups of TGs [85,86]. Furthermore, homo-dimerization of EL enables optimal interaction with and activity against its substrates, and hydrolase activity is regulated by cleavage of pro-protein convertases [87,88]. EL determines HDL plasma levels and the given anti-inflammatory and anti-oxidative properties of the particle itself, inciting interest in EL and its role in the pathologies of various diseases. Pregnancy pathologies, such as fetal growth restriction or diabetes, are associated with impaired hydrolysis of maternal lipoprotein-borne lipids. Accordingly, elevated EL mRNA expression was found in placentae from obese women complicated with GDM and placentae from pregnancies complicated by type 1 diabetes, respectively [35,77,89]. In contrast, EL mRNA showed a significant decrease in IUGR and preeclamptic placentae, which emphasizes an apparent dysregulation of EL in those pregnancy pathologies [27,81]. These results highlight that pregnancy disorders clearly affect the transcription of placental EL, potentially leading to changes in the placental–fetal lipid axis.

### 2.3. Small-Molecule Amidases

Fatty acid amide hydrolase (FAAH) is the only hydrolase with amidase activity that has been described and discussed in female reproductive tissues (Figure 2). This section summarizes current knowledge on the expression and function of this enzyme from the first to third trimesters of pregnancy.

#### Fatty Acid Amide Hydrolase (FAAH)

FAAH is the primary enzyme degrading fatty acid amides, such as the endocannabinoid anandamide (AEA), also called N-acylethanolamine, into AA and ethanolamine. As a member of the amidase signature superfamily of serine hydrolases, FAAH harbors a SSK triad at the catalytically active site. FAAH represents a ~60 kDa integral membrane protein with a strong preference for hydrophobic substrates compared to other members, which are soluble enzymes targeting hydrophilic substrates [15,90]. Human FAAH shows 84% and 82% amino acid sequence similarities to murine and rat FAAH, respectively [91]. FAAH knockout mice displayed significantly elevated endogenous brain levels of AEA, which correlated with increased CB1-dependent anxiolytic and analgesic effects [92]. The potential of AEA for alleviating pain and anxiety has led to significant efforts directed toward the development of FAAH inhibitors as a therapeutic strategy. So far, few FAAH inhibitors have been employed in clinical trials to treat diverse neurological disorders, all tested compounds are centrally active and covalently inactivate FAAH enzyme activity [93,94]. Other attempts to find a selective FAAH inhibitor were abandoned in clinical studies because its application to patients resulted in detrimental side effects [95]. Studies investigating the involvement of the endocannabinoid system in first trimester miscarriage underline the importance of tightly regulated synthesis and degradation of AEA. The examination of placentae from woman either undergoing elective abortion or being affected by spontaneous miscarriage revealed decreased FAAH protein levels between 9 and 12 weeks of gestation in samples of spontaneous miscarriage compared with uncomplicated early pregnancy samples. This was further confirmed by immunohistochemistry showing positive staining for FAAH in the trophoblast layers of gestational age-matched controls compared to undetectable immunoreactivity in early spontaneous miscarriage tissues [96]. Furthermore, FAAH activity and protein levels were significantly lower in the blood lymphocytes of in vitro fertilization-embryo transfer patients who experienced miscarriages compared to those who had become pregnant. This effect was accompanied by increased blood AEA levels [97]. Moreover, genetic or pharmacological depletion of FAAH in mice revealed impaired oviductal embryo transport and development [98]. Importantly, embryo retention in the fallopian tube is a significant cause of ectopic pregnancies in women. An investigation of fallopian tubes obtained from women diagnosed with ectopic pregnancy mirrored the findings mentioned above. The tube epithelium showed reduced FAAH expression and higher AEA levels in ectopic pregnancies compared to luteal phase controls [99]. An additional study by the same group examined plasma endocannabinoid levels and FAAH activity in peripheral blood cell membranes and confirmed increased AEA and reduced FAAH levels in women with ectopic pregnancy compared to healthy controls [100]. These results underline the importance of an appropriate AEA tone for normal implantation. FAAH expression was elucidated in human first-trimester and full-term placental tissues, by immunohistochemistry and the protein was localized to STs, CTs, the decidua, and the endometrium [101,102,103,104]. Additionally, many studies have investigated AEA signaling in the process of parturition. FAAH protein levels and enzymatic activity were decreased in vaginally delivered placentae compared to non-labor cesarean sections, while AEA plasma levels increased with the onset of labor [105,106]. Notably, FAAH gene expression and activity in human lymphocytes is stimulated by physiological serum concentrations of progesterone, which has an immune modulatory function and is known to be critical for normal gestation [107,108]. Therefore, FAAH and its respective lipid mediators might play a role as an immune modulatory trigger to initiate and propagate a normal gestation.

### 2.4. Phospholipase A2 Enzymes

The phospholipase A2 (PLA2) superfamily includes fifteen distinct groups of enzymes containing various subgroups mainly classified by their sequence homology. These enzymes hydrolyze the acyl chain at the sn−2 position of phospholipids, releasing a free fatty acid and a lysophospholipid. The manifold actions of PLA2s originate from the diversity of their substrates, namely the length and grade of saturation of respective fatty acids at the sn−2 position. Consequently, PLA2s are responsible for the inactivation of signaling lipids, such as the platelet activating factor or the release of AA, thereby regulating inflammatory processes [109,110]. More detailed reviews on categorical types, structural characteristics, inhibitors of PLA2 enzymes, and studies of PLA2 in pathological conditions can be found elsewhere [111,112,113]. The PLA2 family members, which were identified in human placental tissue and appear to be involved in the aberrant lipid metabolism of pregnancies complicated by preeclampsia (PE) or obesity, are PLA2 group IIA (PLA2G2A) and PLA2 group V (PLA2G5) (Table 1). Furthermore, mRNA of PLA2 group IV (PLA2G4), PLA2 group VI (PLA2G6), and PLA2 group VII (PLA2G7) were detected in placental tissue biopsies but were not differently regulated in pathological placentae compared to controls (Figure 2) [114,115]. PLA2 enzymes during pregnancy have been extensively reviewed by our group, where we summarized the expression of PLA2 members in human placental tissue and distinct cell types [116]. Besenboeck and others further underline the aberrant dynamics of PLA2 levels in pregnancy pathologies, such as GDM, PE, or preterm delivery [116].

## 3. Concluding Remarks

The maternal lipid metabolism undergoes major and rapid changes across gestation to fulfill the requirements for the development and growth of the fetus [7]. Most of the well-known obstetric pathologies are marked by metabolic alterations and systemically manifested by inflammation, including insulin resistance and elevated proinflammatory cytokine levels [117,118]. Emerging evidence suggests that maternal and placental dyslipidemia is a hallmark of many pregnancy-associated complications, such as PE, preterm delivery, and fetal macrosomia [119,120,121]. Even prior to fertilization, lipid homeostasis has been shown to be crucial for oocyte maturation, as maternal blood TGs, total cholesterol, and low-density lipoprotein levels are negatively correlated with the quality of in vitro fertilized embryos [122,123]. Furthermore, it has been demonstrated that the pre-conceptive metabolic state of the mother determines the chance of suffering from PE or GDM in pregnancy. Women with low HDL cholesterol and high TG levels showed an increased rate of pregnancy complications [124]. In addition, in recent years, many studies have focused on the relation between maternal nutrition and fetal metabolic programming. It has been demonstrated that imbalances in fatty acid supply during intrauterine growth can cause metabolic and endocrine adaptions. These changes may, through developmental programming, lead to an increased risk for metabolic disorders later in adult life, such as obesity and cardiovascular disease [125,126]. Both lipids and fatty acids possess the ability to act as bioactive chemical messengers; thus, it appears that their synthesizing and degrading enzymes tightly regulate their abundance. Although many attempts have already been made to describe alterations of metabolic enzymes linked to certain pathologies (as summarized in Table 1), most of the underlying mechanisms are still unknown. This is due to the fact that, besides the existing descriptive results for bioactive lipid messengers, the field of reproduction lacks activity-based functional studies that could help demonstrate their physiological relevance. Hence, the generation of activity-based protein profiles, precisely elucidating the level of enzyme activity in healthy and diseased tissues, would be a benefit and help obtain deeper insights into the pathomechanisms of certain pregnancy-associated diseases. This high-throughput method allows the identification of proteins and their activities in complex biological samples by utilizing active-site-directed small molecule probes. Furthermore, many studies strongly rely on animal models to approximate as closely as possible the in vivo situation of a diseased state. In fact, inter-species differences in the placental morphology and physiology of gestation hinder direct comparisons between animals and humans. Importantly, recent advances in serine hydrolase inhibitor development will make it possible to undertake functional studies on these enzymes in human tissue. These pharmacological strategies provide a powerful tool to investigate the impact of acute enzyme inhibition on lipid networks in healthy and diseased conditions. Many individual enzymes have already been described in the placenta and female reproductive tissues, but the functions of other members of the serine hydrolase family remain to be elucidated. The application of activity-based methods in the investigation of aberrant hydrolase activities in physiological and pathophysiological conditions of pregnancy may contribute to new hypotheses, early diagnoses, or even new treatment options in the future.

**Table 1 ijms-23-06851-t001:** Expression of hydrolases in human reproductive tissues and their involvement in pregnancy pathologies. Upward- and downward-facing arrows indicate increases or reductions of the respective enzymes in the described pathology.

Lipase	Human Tissue/Cell Type	Detection	Pathology
ATGL	Term Placenta/ST layer, endothelial cells, HBCs and decidua cells [26,27]	mRNA/protein	GDM (↑) [26]
HSL	Term Placenta/ST layer, endothelial cells, HBCs and decidua cells [26,27]	mRNA/protein	PE (↓) [27] GDM (↑) [35]
DAGLα	Term Placenta/CT, ST and BeWo cells [43,44,45]	mRNA/protein	-
DAGLβ	Term Placenta/CT, ST, endothelial cells, HBCs (unpublished data)	mRNA	-
MGL	Endometrium, term placenta/CT and BeWo cells [43,44,45,57]	mRNA/protein	Endometrial carcinoma (↓) [57]
ABHD6/12	Term placental explants [45]	mRNA/protein	-
LAL	Term placenta [35,64]	mRNA	-
LPL	Term placenta/trophoblast cells, endothelial cells, HBCs [72,74,75]	mRNA/protein/activity	GDM/Type 1 Diabetes (↓) [77], IUGR (↑) [79,80,81]
EL	Term placenta, ST, endothelial cells [72,81]	mRNA/protein	Obese GDM/Type 1 Diabetes (↑) [35,77,89] IUGR (↓) [81], PE (↓) [27]
FAAH	Endometrium, first trimester placenta, term placenta/fallopian tube epithelium, CT, ST, endothelial cells, [96,99,101,102,104,127]	mRNA/protein/activity	Miscarriage (↓) [96,97], PE (↓) [127], ectopic pregnancy (↓) [99,100]
Phospholipase A2 enzymes	Term placenta/trophoblast cells, endothelial cells [114,115,116]	mRNA/protein/actvity	PE (↑ PLA2G2A, PLA2G5) [114], Obesity (↑ PLA2G2A, PLA2G5) [115], Preterm delivery (↑ PLA2G2A) [116]

## Figures and Tables

**Figure 1 ijms-23-06851-f001:**
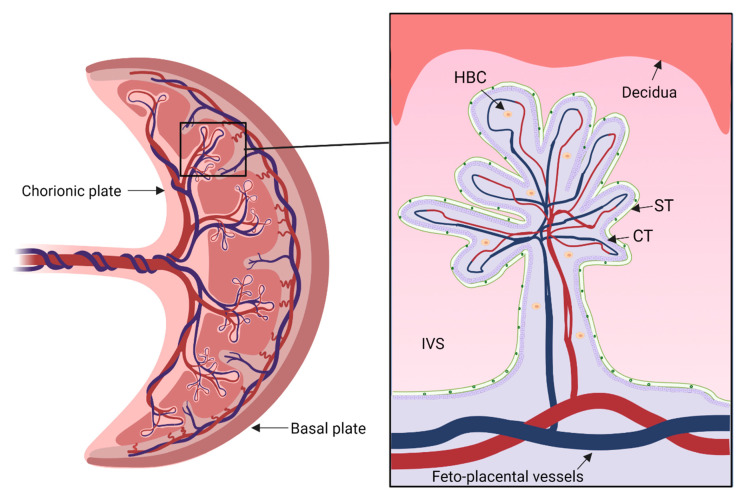
Schematic depiction of the structure of a human placenta. The umbilical cord inserts into the chorionic plate and the basal plate faces the maternal uterus. The expansion shows a term placental villus with the different cell types surrounded by the blood-filled intervillous space (IVS). Hofbauer cell (HBC), cytotrophoblast (CT), syncytiotrophoblast (ST). Created with BioRender.com.

**Figure 2 ijms-23-06851-f002:**
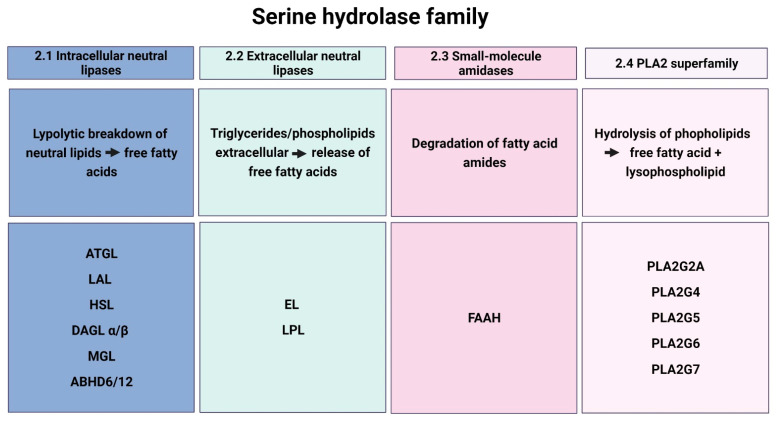
Overview and distribution of described serine hydrolases in the human placenta. Endometrium (E), intervillous space (IVS), syncytiotrophoblast (ST), cytotrophoblast (CT), Hofbauer cell (HBC), endothelial cell (EC). Created with BioRender.com.

## Data Availability

Not applicable.

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
