# Peer review of "Serine Hydrolases in Lipid Homeostasis of the Placenta-Targets for Placental Function?"

_ijms, 2022, doi:10.3390/ijms23126851_

Round 1
Reviewer 1 Report
The authors present a well organized manuscript focused on the expression and importance of serine hydrolases in the placenta.
There are only a few minor points for consideration before the paper is ready for publication.
1. The figures/tables are not labeled correctly or referred to appropriately in the manuscript.
2. The figure of the placental structure would be improved if the intervillous space was labelled too and the location of the decidua eluded to. This would then more accurately reflect the description in section 1.1.
3. if the section numbers (i.e. 2.1-2.4) were added to the serine hydrolase family figure, this would help the reader to immediately understand where the enzyme discussion was to be found in the narrative.
4. The concluding remarks would be improved with suggestions as to which activity based functional studies could be most impactful to the field and briefly mention how the regulation of maternal lipids determines aspects of fetal programming that have lifelong consequences.
Reviewer 2 Report
The authors conducted a comprehensive review of the literature concerning the involvement of serine hydrolases in placental lipid homeostasis. The manuscript is clear and well written.
As minor revision I suggest to briefly discuss the possible variations occuring in pregnancies obtained in donor oocytes cycles.
